# The Urine Metabolome of R6/2 and zQ175DN Huntington’s Disease Mouse Models

**DOI:** 10.3390/metabo13080961

**Published:** 2023-08-18

**Authors:** Roberto Speziale, Camilla Montesano, Giulia Di Pietro, Daniel Oscar Cicero, Vincenzo Summa, Edith Monteagudo, Laura Orsatti

**Affiliations:** 1Experimental Pharmacology Department, IRBM SpA, Via Pontina km 30.600, 00071 Pomezia, Italy; r.speziale@irbm.com; 2Department of Chemistry, Sapienza University of Rome, Piazzale Aldo Moro 5, 00185 Roma, Italy; camilla.montesano@uniroma1.it; 3Department of Chemical Sciences and Technology, University of Rome “Tor Vergata”, Via Cracovia 50, 00133 Roma, Italy; giulia.dipietro.dpg@gmail.com (G.D.P.); cicero@scienze.uniroma2.it (D.O.C.); 4Department of Pharmacy, University of Napoli “Federico II”, Corso Umberto I 40, 80138 Napoli, Italy; vincenzo.summa@unina.it; 5CHDI Management/CHDI Foundation, 6080 Center Drive, Los Angeles, CA 90045, USA; edith.monteagudo@chdifoundation.org

**Keywords:** urine metabolome, mass spectrometry, R6/2 mice, zQ175DN mice, Huntington’s disease

## Abstract

Huntington’s disease (HD) is caused by the expansion of a polyglutamine (polyQ)-encoding tract in exon 1 of the huntingtin gene to greater than 35 CAG repeats. It typically has a disease course lasting 15–20 years, and there are currently no disease-modifying therapies available. Thus, there is a need for faithful mouse models of HD to use in preclinical studies of disease mechanisms, target validation, and therapeutic compound testing. A large variety of mouse models of HD were generated, none of which fully recapitulate human disease, complicating the selection of appropriate models for preclinical studies. Here, we present the urinary liquid chromatography–high-resolution mass spectrometry analysis employed to identify metabolic alterations in transgenic R6/2 and zQ175DN knock-in mice. In R6/2 mice, the perturbation of the corticosterone metabolism and the accumulation of pyrraline, indicative of the development of insulin resistance and the impairment of pheromone excretion, were observed. Differently from R6/2, zQ175DN mice showed the accumulation of oxidative stress metabolites. Both genotypes showed alterations in the tryptophan metabolism. This approach aims to improve our understanding of the molecular mechanisms involved in HD neuropathology, facilitating the selection of appropriate mouse models for preclinical studies. It also aims to identify potential biomarkers specific to HD.

## 1. Introduction

Huntington’s disease (HD) is caused by a dominant mutation in the huntingtin (*HTT*) gene, which consists of an abnormal expansion of a gene’s CAG repeat [1,2,3]. The gene mutation codes for an abnormally long glutamine stretch (polyQ) in the huntingtin (HTT) N-terminal sequence (exon 1), leading to the devastating neurodegenerative disease. An individual with CAG expansion exceeding 35 repeats in one copy of the *HTT* gene will certainly suffer from HD. It has been demonstrated that the age of onset is inversely proportional to the length of the CAG expansion, and the disease is less pervasive when the CAG repeat length is shorter [4]. Since the discovery of the molecular basis of HD [5], there has been intense research aimed at understanding the fundamental principles underlying the mechanism of HD. An important area of research is the establishment of accurate HD mouse models to use in preclinical studies of disease mechanisms, target validation, and therapeutic compound testing. This line of investigation has resulted in the production of a large variety of HD mouse models showing variable symptom severity and disease onset depending on the genotype [6,7,8], none of which fully recapitulate the human disease, complicating the selection of an appropriate model. Two widely used HD mouse models are R6/2 and zQ175DN. R6/2 was among the first transgenic mouse models that were developed [9,10] and expresses exon 1 of the human HD gene with approximately 110 and 160 CAG repeats. The levels of transgene expression, driven by the human huntingtin promoter, are approximately 75% of endogenous huntingtin [11]. It exhibits the early onset (6–8 weeks) of HD symptoms, including muscular atrophy, decreased body weight, reduced bone mineral density, abdominal fat accumulation, insulin resistance, and cognitive deficits, as well as motor and behavioral abnormalities that parallel HD in humans. A progressive alteration in the hypothalamic–pituitary–adrenal axis, reminiscent of a Cushing-like syndrome, has also been reported [12,13,14]. The R6/2 mice have the most widespread distribution of HTT inclusions in the brain and develop symptoms most rapidly compared with all other HD mouse models. The zQ175DN heterozygous knock-in (KI) mice have the mouse HTT exon 1 replaced by the human HTT exon 1 sequence with ~180–220 CAG repeats. Furthermore, the neo-cassette of these KI mice was deleted (DN) [15]. It is heterozygous for one wild-type *HTT* allele and one CAG-expanded allele and compared with the R6/2 model shows lower variability in gene expression levels and avoids random transgene integration. Motor symptoms start from 3 to 4 months, while brain atrophy initiates from 8 months. Overall, the zQ175DN-KI model shows a significantly slower progression of the disease than R6/2 mice. In the present work, liquid chromatography (HILIC and RP) coupled with high-resolution mass spectrometry (LC-HRMS)-based metabolite profiling was used in conjunction with univariate and multivariate statistical analyses to elucidate urine metabolic alterations in two widely used HD mouse models: R6/2 and zQ175DN (Figure 1).

Recent advances in separation techniques and high-resolution mass spectrometry (HRMS), combined with chemometrics, have provided new attractive tools for metabolomics research [16,17,18]. Urine is an accessible biological fluid, easy to collect, and rich in metabolites, and several reports demonstrated that urinary metabolic alterations could reflect biochemical activity implicated in the pathogenesis of a genetic disease. We used this approach to understand if the urinary metabolome can distinguish different HD mouse genotypes and thus help to employ this information to select the appropriate model for preclinical studies. According to our research, R6/2 mice had higher levels of corticosterone metabolites than wild-type mice, suggesting that the corticosterone metabolic pathway is one of the most dysregulated pathways in this model. Moreover, an increase in pyrraline, considered a diagnostic marker of diabetes, was observed. Xanthurenic acid and its glucuronidated derivative were decreased, suggesting alterations in tryptophan metabolism. Finally, the secretion of the pheromone, 2-(sec-butyl)-4,5-dihydrothiazole, appeared to be impaired due to the profound pathological changes in the R6/2 brain. In the zQ175DN mice, an increase in oxidative stress biomarkers was noticed together with an alteration in the tryptophan metabolic pathway. This approach aims to understand how different models reflect disease phenotypes and mechanisms, enabling the delineation of altered biochemical pathways associated with genotypes and revealing disrupted molecular mechanisms. The final goal is the selection of appropriate models for preclinical studies and the identification of biomarkers specific to HD or the underlying neurodegenerative process.

## 2. Materials and Methods

### 2.1. Standards and Reagents

All the solvents and reagents used were LC-MS-grade. Water (H_2_O), acetonitrile (ACN), formic acid (FA), and ammonium formate (CAS 540-69-2) were obtained from Sigma-Aldrich (St. Louis, MO, USA). The SIL internal standard ^13^C^15^N_2_-8-hydroxy-2′-deoxyguanosine (^13^C^15^N_2_-8-OH-dG) was obtained from Toronto Research Chemicals (Toronto, ON Canada); ^15^N_4_-hypoxanthine (^15^N_4_-Hyp), L-tyrosine-(phenyl-d_4_) (d_4_-L-Tyr) ^15^N_4_-inosine (^15^N_4_-I) were purchased from Cambridge Isotope Laboratories, Inc. (Tewksbury, MA USA); L-kynurenine sulfate: H_2_O (ring-d_4_, 3,3-d_2_) (d_6_-KYN) was obtained from Cambridge Isotope Laboratories, Inc. (Andover, MA, USA); anthranilic acid-ring-^13^C_6_ (^13^C_6_-AA) was obtained from Sigma-Aldrich (Schnelldorf, Germany); ^15^N,^13^C_2_-3-Hydroxy-DL-kynurenine (^15^N-^13^C_2_-OH-KYN) was obtained from AMRI (Albany, NY, USA); and 3,5,6,7,8-d_5_-kynurenic acid (d_5_-KYNA) was obtained from Qmx Laboratories (Thaxted, UK).

### 2.2. SIL Stock and Working Solution Preparation

Stock solutions were prepared from the independent weight of compounds and stored at −20 °C. First, d_6_-KYN and ^13^C_6_-AA were prepared in water/DMSO (1/1, *v*/*v*) at 1.5 and 5.0 mg/mL, respectively. ^15^N^13^C_2_-OH-KYN were prepared in water/DMSO (1/19, *v*/*v*) at 2.0 mg/mL. ^13^C^15^N_2_-8-OH-dG, ^15^N_4_-Hyp, d_4_-L-Tyr, and ^15^N_4_-Inosine were prepared in water at 1.0 mg/mL.

Internal standard working solutions (ISWS) were prepared by adding appropriate volumes of the stock solutions to 50 mL of ultrapure water (ISWS-A) and acetonitrile (ISWS-B), respectively, to reach a final concentration of 200 ng/mL for all the standards. The solutions were maintained at 4 °C and freshly prepared every week.

### 2.3. Animals and Study Design

#### 2.3.1. R6/2 Mice

Transgenic R6/2 mice expressing the exon 1 of the human HD gene as an early-onset HD model were used in this study. Animals were maintained in the Charles River Finland Facility. Transgenic R6/2 males were bred with nontransgenic C57BL/6J females producing mixed B6CBA/J backgrounds crossed to C57BL/6J animals. CAG size was 120 ± 10. After animal genotyping, transgenic and nontransgenic animals were group-housed by genotype. Non-Tg mice were used as controls.

#### 2.3.2. zQ175DN KI Mice

The zQ175DN heterozygous knock-in mouse (B6JzQ175DN KI, JAX stock # 029928) has the mouse HTT exon 1 replaced by the human HTT exon 1 sequence, with a ~190 CAG repeats tract. This HD model has the floxed neo-cassette removed resulting in a ~2-fold increase in mutant HTT protein in the brain of zQ175DN mice. These mice exhibit an improved reproduction of HD phenotype. Animals were maintained in the Charles River Finland Facility. Male mice were used. Wild-type mice from the colony were used as controls.

#### 2.3.3. Study Design and Urine Collection

All animal experiments were performed at Charles River Laboratories (Kuopio, Finland) and were carried out according to the National Institute of Health (NIH) guidelines for the care and use of laboratory animals and approved by the National Animal Experiment Board. Mice were prescreened, and those with abnormally high plasma bile acid levels were excluded from the study. A total of 24 (8 Tg and 16 non-Tg) and 38 (18 knock-in and 20 wild-type) male mice were used. For R6/2, Tg and wild-type mice, 5-week-old and 14–15-week-old mice were used (*n* = 3 Tg and *n* = 4 wild-type for 5 weeks old, respectively; *n* = 5 Tg and *n* = 12 wild-type for 14–15 weeks old, respectively). For zQ175DN and respective wild-type mice, 4-, 12-, and 15.7-month-old mice were used (*n* = 2 knock-in and *n* = 4 wild-type for 4-month-old mice, respectively; *n* = 7 knock-in and *n* = 6 wild-type for 12-month-old mice, respectively; *n* = 9 knock-in and *n* = 10 wild-type for 15.7-month-old mice, respectively). Urine was collected daily into separate Eppendorf tubes by holding each mouse between the thumb and index finger. This procedure was repeated over several days until 350 µL of urine was collected. Urine samples from each mouse were combined into one sample and centrifuged (5 min, 400× *g*, 4 °C) to remove precipitates. The clear supernatant was collected and stored at −80 °C.

### 2.4. Mouse Urine Normalization via Specific Gravity

A digital refractometer (Atago UG-α; Tokyo, Japan) was used to measure the specific gravity (SG), with a urinary range (1.000–1.060) and 0.001 resolution. Urines were thawed at room temperature, sonicated in a water bath for 10 min, and then centrifuged (13,000× *g*, 10 min). Briefly, 100 µL aliquot was used to measure SG; after calibration with LC-MS grade water, the samples were placed upon the lens of the refractometer. The samples were then split into two aliquots. Urinary metabolite levels were normalized by diluting each aliquot with water or a mixture of water and acetonitrile in variable amounts for RP and HILIC analysis, respectively. Dilutions were performed to achieve a common SG for all samples.

### 2.5. Urine Sample Preparation

For RP and HILIC analyses, all samples were diluted three times with ISWS-A and ISWS-B, respectively. After vortexing, samples were centrifuged (13,000× *g*, 10 min), and the supernatant (350 μL) was transferred to a 96-well plate for LC-HRMS analyses. Samples from R6/2 and wild-type mice were analyzed separately from the zQ175DN and wild-type ones. Samples were randomized in both experimental runs.

### 2.6. The Preparation and Analysis of Quality Control Samples and Blanks

Two sets of quality control (QC) samples were prepared. These included (1) pooled QCs prepared by mixing equal volumes (5 μL) from each sample previously normalized for the SG and (2) dilution QCs prepared by diluting 2, 4, and 8 folds the pooled QCs with water. All QCs were further diluted 3 folds with ISWS-A for RP analysis, or with ISWS-B for HILIC analysis. To obtain stable retention times and MS responses, a total of 20 pooled QC samples were injected at the beginning of the run to condition the LC-HRMS system. To correct the intra-batch signal drift, the pooled QCs were injected into every six true samples (*n* = 8 in total). To verify the linear response of the MS signal, the dilution QCs were analyzed four times and regularly added to the sample list. Blank samples consisting of LC-MS-grade water for RP analysis and ACN–water 80:20 (*v*/*v*) for HILIC analysis were injected (*n* = 3) at the beginning of the batch to collect a background signal to exclude from the dataset. The pooled QC samples analyzed within the batch were used to assess the method’s within-batch precision, as well as the technical replicate precision.

### 2.7. UHPLC–High-Resolution Mass Spectrometry Analysis

#### 2.7.1. HILIC and Reverse-Phase Chromatography

UHPLC–high-resolution mass spectrometry (UHPLC–HRMS) analysis was performed as described in Petrella G. et al. [19]. Briefly, an Orbitrap QExactive™ mass spectrometer coupled with an Ultimate 3000™ liquid chromatographic system (Thermo Scientific™, Waltham, MA, USA) was used, equipped with a heated electrospray ionization (HESI-II) source operated in negative and positive ion mode. Reverse-phase (RP) chromatographic separation was accomplished using an HSS-T3 (Waters, Milford, MA, USA) column, 100 Å, 1.7 μm, 2.1 × 100 mm and with the temperature set at 35 °C. The mobile phases were 0.1% formic acid in water (mobile phase A) and 0.1% formic acid in acetonitrile (mobile phase B). The following gradient profile was used: mobile phase B started at 0%, increased linearly to 10% in 6 min, and to 35% in 2 min. Phase B further increased to 98% in 2 min and was kept stable for 1.5 min. Then, in 1.5 min, it was returned to 0% and kept stable for 3 min for the column’s final equilibration. The flow rate was 0.3 mL/min from 0 to 8.00 min, increased to 0.4 mL/min from 8.00 to 12.0 min for column washing, and brought back to 0.3 mL/min from 12.0 to 15.0 min. The injection volume was 2 µL. HILIC chromatographic separation was achieved using a BEH-HILIC (Waters, Milford, MA, USA) column, 130 Å, 1.7 μm, 2.1 × 100 mm at a temperature of 35 °C. The mobile phases used were 20 mM ammonium formate + 0.1% formic acid at pH 3.7 (mobile phase A) and acetonitrile (mobile phase B). The gradient started from 5% mobile phase B, ramped to 35% over 8.5 min, followed by an increase to 50% for 1 min. It was kept stable for 1.5 min and finally decreased to 5% B in 0.5 min and stayed at 5% B for 3.5 min for column re-equilibration. The flow rate used was 0.3 mL/min, and the injection volume was 2 µL. Samples were kept in the autosampler at 8 °C during the LC-HRMS analysis.

#### 2.7.2. Mass Spectrometry

Mass spectra were acquired in both positive and negative ion modes. The HESI parameters were 3.20 kV (pos)/−3.20 kV (neg) electrospray voltage, 280 °C heated capillary temperature, 50 (pos)/−50 (neg) S-lens RF level, sheath gas (N2) flow 50 a.u., auxiliary gas (N2) flow 10 a.u., and gas temperature of 300 °C. A full-MS/dd-MSMS method was used consisting of a full scan (FS) and three data-dependent MS/MS scans on the three most intense ions in the full scan. For FS MS, the acquisition range was from *m*/*z* 60 to 900, and the resolution was 70,000 FWHM (at *m*/*z* 200). For MS/MS, the resolution was 17,500 FWHM (at *m*/*z* 200), and parent ion fragmentation was performed using stepped normalized collision energy (NCE) of 20 and 50 (eV). The intensity threshold was set to 2.0 × 10^4^, dynamic exclusion was 2.0 s, and the underfill ratio was 5%. The MS/MS spectra of features of interest, if not available from the dd acquisition, were acquired in a separate sample injection. All data were acquired in profile mode using Xcalibur™ 3.1.66.10.

The QExactive™ mass spectrometer was calibrated for positive and negative modes before sample analysis using the calibration solution provided by the manufacturer (Pierce LTQ ESI Positive Calibration Solution and Pierce LTQ ESI Negative Calibration Solution). For the mass calibration of the instrument, a custom list that included lower masses than the default calibration provided with the instrument was used to ensure that accurate masses were detected at low molecular weights.

### 2.8. Data Analysis

#### 2.8.1. Raw Data Processing Using Compound Discoverer

The raw files obtained in both positive and negative ion modes were processed separately using Compound Discoverer^TM^ 2.0 (Thermo Scientific^TM^, Waltham, MA, USA). Four output tables (RP pos, RP neg, HILIC pos, and HILIC neg) were created for each genotype, which contained *m*/*z*, retention time, and peak intensity for all the samples analyzed. To align retention time, detect components, predict elemental composition, and fill gaps, a nontargeted metabolomic workflow was utilized. The workflow tree consisted of nodes such as input files, selecting spectra, aligning retention times, detecting unknown compounds, grouping unknown compounds, filling gaps, normalizing areas, and marking background compounds. The raw files were aligned with an adaptive curve setting with 5 ppm mass tolerance and 0.4 min retention time shift. The unknown compounds were detected with 5 ppm mass tolerance, 3 signal-to-noise ratio, 30% of relative intensity tolerance for isotope search, and 10,000 minimum peak intensity, and then grouped with 5 ppm mass and 0.3 min retention time tolerances. In the pretreatment step, a procedural blank sample was used to remove the noise and subtract the background. Peaks with less than a three-fold increase were matched to blank samples, and those detected in less than 50% of QCs and where the relative standard deviation (%RSD) of the QCs was greater than 50% were removed from the list. To balance the differences in intensities that may have arisen from instrument instability, a normalized area was provided across all samples for each detected metabolic feature, by normalizing to the QC samples periodically analyzed (pooled QC).

#### 2.8.2. Raw Data Filtration

Two filtration strategies were serially applied to reduce the final number of hits undergoing statistical evaluation. A dilution–filtration approach was first used by correlating the signal intensity obtained for the diluted QCs to the dilution factor. For each detected metabolic feature, the correlation coefficient (R^2^) was calculated using an Excel add-in, and a cutoff of 0.8 and a %CV for pooled QC below or equal to 30% (*n* = 8) was applied. Data were further filtered using an in-house MATLAB script that allowed the clusterization of the ions generated from the same parent (adducts, dimers, fragments, and isotopes) based on Pearson’s correlation of their intensity.

#### 2.8.3. Multivariate Statistical Analysis (SIMCA)

After filtration, the four datasets for each genotype (RP pos, RP neg, HILIC pos, and HILIC neg) were combined, to generate the output table, which was exported to Simca-P 14 (Umetrics, Umea, Sweden) for multivariate analysis via principal component analysis (PCA) to identify outliers (samples that are extremely different from the rest of the dataset). Afterward, to identify the differences between specific sample groups, orthogonal partial least square discriminant analysis (OPLS-DA) as a supervised multivariate approach was used to study the contribution of the variables in group separation. Moreover, to confirm the goodness of fit of the supervised multivariate approach, permutation plots were drawn. This analysis was able to confirm that the model represented the better way to separate the groups in our dataset. The S plots were used to highlight the ions with the greatest influence on the separation between groups. R^2^ and Q^2^ were used to assess the overall model performance and its ability to predict class membership.

The hit selection mode from the S plot was set using a handmade MATLAB script. Metabolic features showing VIP > 2.0 and *p*-value < 0.01 (*t*-test, Wilcoxon–Mann–Whitney) were selected. To highlight the most interesting hits, P [1] and P (corr) ad hoc values were chosen.

#### 2.8.4. Putative Metabolite Identification

Putative metabolite identifications were performed by matching accurate mass data and isotope patterns with mass spectral and compound libraries like PubChem (http://pubchem.ncbi.nlm.nih.gov/), ChemSpider (http://chemspider.com/), HMDB [20] (http://www.hmdb.ca/), KEGG Pathway mapping (https://www.genome.jp/kegg/pathway, [21], and Lipid Maps (https://www.lipidmaps.org) online databases. In the case of presumable positive findings, the MS/MS spectra acquired in a second injection were compared with mass spectral libraries like mzCloud™ (https://www.mzcloud.org/) and Metlin (https://metlin.scripps.edu/ [22]. The proposed structures were inspected manually.

#### 2.8.5. Metabolite Confirmation with Synthetic Standards

When available, a synthetic standard of a putative metabolite was used to confirm the identification. A direct match of the metabolite’s mass, MS/MS spectra, and RT to that of an authentic standard under identical analytical conditions was provided for the full confirmation of structural identification.

## 3. Results

### 3.1. Mouse Urine Samples’ Normalization via Specific Gravity

In urine, solute concentrations vary widely depending on water consumption and other physiological factors. Thus, a normalization strategy is required for the quantitative comparison of metabolites. Different normalization techniques, which can be used either pre-acquisition (preventive) or post-acquisition (curative), are commonly applied in LC-MS studies [23]. In this work, a physiological pre-acquisition technique, i.e., normalization to SG, was selected to improve the analytical conditions by levelling the urine concentrations and preventing analytical variation [24]. SG of the urine samples was measured using a digital refractometer, and each sample was diluted to reach 1.0073. Furthermore, all samples prepared for HILIC analysis had the same final amount of organic solvent (40% acetonitrile). The applied dilution factor varied from 3.6 to 8.6 for the less and the most concentrated sample, respectively.

### 3.2. Analytical Quality Assurance

Considering the high number of samples analyzed within the study, the robust quality assurance of the LC-HRMS data was crucial. To ensure a high level of data confidence and reliability, the pooled QC samples were injected every six samples to provide signal drift correction for each detected metabolic feature; the signal correction was performed using Compound Discoverer™ software, which provided normalized areas. Seven stable isotopically labeled (SIL) internal standards (ISWS-A and ISWS-B) were spiked into all samples and QCs, to be present at the same concentration; the samples were quantitatively processed before data analysis, and IS peak area was integrated to monitor signal variability over time due to sample processing or MS instrument response. IS peak area and RT were used to create a “control chart” in Excel. Median area values were calculated, and it was verified that the area of the internal standard and RT were within the median ±2σ for each analyzed sample. No outliers were found in the analyzed batches, and all samples were then maintained for the following statistical evaluation.

### 3.3. The Acquisition and Filtering of R6/2 and zQ175DN Datasets

To detect analytes with a wide range of physicochemical properties that maximize metabolome coverage, sample analysis was performed using hydrophilic interaction (HILIC) and reverse-phase (RP) chromatography in positive and negative polarity modes. zQ175DN and R6/2 samples were analyzed and processed separately, as shown in Figure 2.

After raw data acquisition, peak picking, and alignment, an intensity normalization based on the pooled QC signals was performed using Compound Discoverer™ software. The workflow tree displaying the selected data processing nodes and the associated connections is shown in Figure 3, together with the settings of the most critical nodes.

For R6/2 analysis, the original HILIC and RP datasets (positive and negative combined) included 450,281 and 245,075 metabolic features, respectively (total 695,356 hits), while for zQ175DN, 534,103 (HILIC) and 542,021 (RP) metabolic features were detected (total 1,076,124 hits). Thus, two filtering processes were serially applied to reduce redundancy and noise. The initial data reduction method sought to remove the noisy features from the dataset; the rationale of this dilution–filtration strategy was that a linear relationship between MS signal intensity and the metabolite concentration should exist for the features that are useful to characterize metabolic variations in the biological samples [25]. The diluted QCs were prepared to distinguish between informative and uninformative signals; for each metabolic feature, the correlation factor between the peak area and the dilution factor was calculated. Only the features that showed a coefficient R^2^ ≥ 0.8 and a %CV ≤ 30% (*n* = 8) were maintained in the dataset, while the others were filtered out. Uninformative signals are likely to occur in cases such as ionization competition, saturation effects, contamination from the solvents, column leaks, etc. Nearly 98% of zQ175DN and 97% of R6/2 features were filtered out at this stage. The clustering of multiple ion types and fragments was then attempted using Pearson’s correlation of signal intensities [26]. To this aim, a MATLAB script was created in-house. In this script, HILIC and RP features were processed separately; features were listed based on their retention time (RT), and for the ones that shared the same RT (±0.02 min), the peak intensities of each sample were correlated according to Pearson’s correlation. Consequently, features with the same RT, an r > 0.9, and *p* < 0.05 were grouped. Inside each group, only the hit with the highest intensity was maintained for the following statistical analysis. New datasets were built combining the noncorrelated hits with the hits showing the highest intensity among the correlated ones. The homemade filter produced a 20% and 32% hit reduction for zQ175DN and R6/2, respectively. The final filtered datasets containing 14,712 features for R6/2 and 16,097 for zQ175DN were imported to SIMCA-P 14 for statistical analysis.

### 3.4. Multivariate Analysis

Initially, principal components analysis (PCA) was used as an unsupervised multivariate analysis method to visualize the data. The PCA of the filtered datasets (RP plus HILIC) acquired for R6/2 (5 and 14–15 weeks old) and zQ175DN mice (4, 12, and 15.7 months old) urine samples are shown in Figure 4 and Figure 5, respectively.

For the R6/2 HD model, a separation between 14–15-week-old Tg mice and the other 5- and 14–15-week-old groups was visible. A 5-week-old wild-type sample behaved as an outlier and was thus excluded from the dataset for OPLS-DA analysis. For zQ175DN and wild-type mice, the PCA score plot showed moderate separation between the groups by age, independent from the genotype, as well as separation between zQ175DN mice at 15.7 months and wild-type groups.

OPLS-DA was then used to discriminate between R6/2 and wild-type mice at 14–15 weeks (Figure 6A) and between 12- and 15.7-month-old zQ175DN and wild-type mice at 4, 12, and 15.7 months (Figure 7A). Since only two samples were representing the knock-in mice at 4 months, they were excluded from the dataset.

Both HD mouse models’ OPLS-DAs performed well in differentiating between the corresponding age-matched control groups. R2Y and Q2X values reflecting the goodness of fit and the predictive power of the models were 0.99 and 0.97 for the R6/2 versus wild-type animals (CV ANOVA = 7.08 × 10^−9^ and 0.94 and 0.78 for the knock-in versus wild-type model (CV ANOVA = 5.9 × 10^−10^), respectively. The permutation plots (*n* = 999) further showed the goodness of fit of the models, confirming that they represented the best way to separate the groups in our datasets (Figure 6B and Figure 7B).

To identify the features contributing to the separation between the groups for the two HD models (R6/2 versus wild-type mice and zQ175DN knock-in versus wild-type mice), the S plots from OPLS-DA analysis were generated, and *p*-values and fold changes were calculated from the nonparametric Wilcoxon–Mann–Whitney test (one-way ANOVA). The metabolites with variable importance in the projection (VIP) > 2.0 from OPLS-DA and significance level *p*-value < 0.01 from nonparametric Wilcoxon–Mann–Whitney test (P [1] > 0.03 and P [1] < −0.0235, −0.05 < P (corr) < 0.05) were considered discriminative and are highlighted as red circles in the S plot. Figure 8 shows the S plot of (A) R6/2 versus wild-type mice and (B) zQ175DN versus wild-type mice.

To evaluate data robustness, PCA and OPLS-DA multivariate analyses of QC pool samples (technical replicates) were performed, resulting in score plots for both datasets with good clustering of samples, indicating good-quality data.

### 3.5. Urine Metabolite Alterations in 3-Month-Old R6/2 versus Wild-Type Mice

The features considered responsible for the separation of R6/2 versus age-matched wild-type mice groups are listed in Table 1.

A total of 18 altered metabolites were recognized. A large portion of them (6/18) were putatively identified as corticosterone metabolites were found to be increased in urine samples of R6/2 mice, suggesting corticosterone dysregulation. An increase in serum and urine corticosterone levels has already been observed in R6/2 mice and is associated with progressive alterations in the hypothalamic–pituitary–adrenal (HPA) axis, reminiscent of a Cushing-like syndrome [12]. As suggested by Dufour et al. [14], elevated corticosterone levels can contribute to the development of symptomology in transgenic HD mice. Interestingly, increased urinary cortisol levels (the homolog of corticosterone in humans), correlated with disease progression, were also found in humans [27], suggesting that a perturbation of the HPA axis may also be of clinical relevance in human diseases. Another interesting metabolite that was observed as altered was pyrraline (and des-O-pyralline), which was highly elevated in R6/2 mice urines compared with wild-type ones. Pyrraline is an advanced glycation end product formed via a nonenzymatic reaction of glucose with a lysine amino group on proteins and is considered a biochemical marker of diabetes [28,29]. This finding is consistent with the progressive development of diabetes by R6/2 mice due to insulin resistance, as reported by Björkqvist et al. [30]. In vivo, pyrraline can further react with other amino acids’ nucleophiles and form protein crosslinks that are normally observed in diabetic patients. Interestingly, pyrraline was found in neurofibrillary tangles and senile plaques of Alzheimer’s disease brains, and it was proposed to be responsible for the insolubility and resistance to the proteases of the lesions [31].

An additional metabolite detected in non-Tg mice but absent in Tg mice urine was 2-(sec-butyl)-4,5-dihydrothiazole, which is a volatile pheromone secreted by rodents such as mice and rats and excreted in urine [32,33]. This pheromone promotes aggression amongst males while inducing synchronized estrus in females. Its absence in the urine of Tg mice is probably a consequence of the profound alterations occurring in the mouse brain during disease development. Xanthurenic acid and its glucuronidated derivative were decreased, suggesting alterations in tryptophan metabolism.

Three other altered metabolites were detected, but their identities remained unknown. They were all present in non-Tg but absent in Tg mice urine. Further altered metabolites tentatively assigned as N-[(1E)-3-methyl-1,3-butadien-1-yl]-L-cysteine (reduced by 983 folds) and N-acetyl-aspartyl-lysine (increased by 44 folds) were found. The latter tentatively assigned based on the exact mass and MS/MS spectrum has already been detected in human urine as the nonacetylated form [34]. Based on the presented data, it was possible to depict the major metabolic alterations related to HD development in the R6/2 mouse model. In summary, the major altered pathways were corticosterone dysregulation, the nonenzymatic glycation of proteins due to insulin resistance, Trp metabolism, and the inhibition of pheromone excretion.

### 3.6. Urine Metabolite Alterations in zQ175DN versus Age-Matched Wild-Type Mice

A total of 14 differential metabolites (Table 2) were responsible for the separation observed in the OPLS-DA model between 12- and 15.7-month-old zQ175DN knock-in and wild-type control mice at 4, 12, and 15.7 months of age.

A large portion of the altered metabolites, including N-acethylglutathione, thiodiglycolic acid (TDGA), carnosine–propanal–aspartic, S-2-carboxyethyl-L-cysteine/2-((2-Hydroxyethyl)amino)-3-mercapto-4-oxobutanoic acid, were related to oxidative stress. TDGA is a urine metabolite that increases when the redox equilibria of an organism are changed [35]. Carnosine–propanal is formed via the reaction of the endogenous dipeptide carnosine with the reactive aldehyde acrolein, which is produced via the oxidation of unsaturated lipids and accumulates in tissues during inflammation (detoxification). Carnosine–propanal is excreted in the urine of mice and adult humans. Carnosine is highly concentrated in muscle and brain tissues and can scavenge ROS and RNS. Subsequently, carnosine–propanal forms covalent adducts with nucleophilic amino acids, leading to the generation of carnosylated proteins. A metabolite increased by 3.3 folds in zQ175DN mice compared with controls was indole-3-pyruvic acid glucuronide, which reflects an alteration in the tryptophan metabolic pathway. It is formed via the transamination of L-tryptophan by tryptophan-2-oxoglutarate aminotransferase [36]. The enolic form of the primary reaction product is indole-3-pyruvic acid. Indole-3-pyruvic acid is highly susceptible to reactive oxygen species (ROS) and readily undergoes pyrrole ring cleavage through interaction with oxygen intermediaries. The transiently formed product then spontaneously cyclizes to generate kynurenic acid (KYNA). Other perturbed metabolites not related to specific pathways were trimethylaminoacetone; deamidated glutathione, which is considered a waste compound; the dipeptide gamma-glutamyltaurine; and a fatty amide N-methyl hexanamide. The latter was already found in rat urine after the dosing of acylamide in a toxicokinetic study. Other altered metabolites (four in total) were not identified. In summary, for zQ175DN mice, urine metabolite profiling revealed an increase in oxidative stress and perturbation of the tryptophan metabolism pathway. No evidence of corticosterone dysregulation or diabetes development was found, revealing a substantial metabolic difference between the two genotypes.

## 4. Discussion

In this study, we used a noninvasive HRMS-based untargeted metabolomic approach to delineate the metabolic pathways altered in the urine of two widely used HD models, R6/2 and zQ175DN knock-in mice, and the underlying biochemical changes involved in disease progression.

Urine samples were collected from the different genotypes and aged-matched wild-type animals. A preliminary normalization using SG was performed to account for the variation in urine concentration due to water consumption. After normalization, the samples were simply diluted and analyzed using two orthogonal chromatographic columns (HILIC and RP). Then, HR-MS data were processed using the dedicated software for data alignment, normalization, and filtering, generating a dataset for each genotype for unsupervised and supervised statistical analysis. The separation between control and HD groups was obtained for both genotypes, and the metabolic features responsible for group separation were identified at level 1 or 2 as defined by the Metabolomics Standards Initiative. Not surprisingly, the altered metabolic pathways were different for the two HD models, consistent with genetic and phenotypic differences. R6/2 is a transgenic model characterized by early disease onset and severe phenotype. Motor and cognitive deficits already start at 6–8 weeks of age, associated with muscular atrophy, decreased body weight, reduced bone mineral density, and abdominal fat accumulation. Insulin resistance and progressive alterations in the hypothalamic–pituitary–adrenal axis are also present. zQ175DN mouse is a less severe model. It is heterozygous for one wild-type HTT allele and one CAG-expanded allele and shows lower variability in gene expression levels. Motor symptoms start from 3 to 4 months of age, and brain atrophy initiates from 8 months. It also shows a significantly slower progression of the disease compared with R6/2 mice. We first compared the urine metabolome of each disease model with its respective age-matched wild-type control and identified the distinctive metabolic features associated with each genotype. We found that in R6/2 mice, the most altered metabolic pathways were corticosterone dysregulation and nonenzymatic glycation, recapitulating the alteration in the hypothalamic–pituitary–adrenal axis and the insulin resistance developed by this model. Also, an alteration in Trp metabolism and the impairment of pheromone secretion was found, underlining the profound biological changes occurring in the mouse brain during disease progression. In zQ175DN knock-in mice, perturbed pathways were mostly different from R6/2 mice. In this model, an increase in oxidative stress metabolites together with an alteration in the tryptophan metabolic pathway, suggested by the increased urine excretion of indole-3-pyruvic acid glucuronide, were the major perturbed pathways.

This is the first untargeted analysis of urine metabolomic changes generated by HD progression in mice. In this study, urine metabolomic analysis revealed different pathways altered in two mouse models of HD widely used in preclinical studies. We demonstrated for the first time that urinary metabolite levels can specifically correlate with mouse HD genotypes and with disease progression. In the future, we will expand our study to other HD mouse genotypes to further validate our platform.

This approach attempts to improve our understanding of the molecular mechanisms involved in HD neuropathology in R6/2 and zQ175DN mice, facilitating the selection of appropriate models for preclinical studies. It also aims to identify potential biomarkers specific to HD or the underlying neurodegenerative process.

Our approach is noninvasive and could be used to profile the urine metabolome from any HD preclinical model and clinical sample. The study design was carefully evaluated in order to reduce the confounding effects induced by sample collection, handling, and storage, contributing to the acquisition of high-quality data. Diet could also be a confounder when it cannot be controlled. Nevertheless, the process of the identification of unknown altered metabolites is the most challenging and time-consuming part of an untargeted metabolomic study, especially when an authentic standard is not available, and the metabolite is not present in the spectral database. It is essential to understand the biological changes contributing to alterations in metabolic features. In this case, if sufficient sample volume is available and enrichment is possible, MS in synergy with NMR analysis could help elucidate metabolite structure.

## 5. Conclusions

In this study, an untargeted metabolomic approach based on HILIC/RP-HRMS was used to profile urine samples from two HD mouse models (R6/2 and zQ175DN). R6/2, a transgenic model, exhibits a more severe phenotype than the knock-in zQ175DN model. Urine samples collected from the two HD mouse models and their respective age-matched wild-type animals were compared. In both models, the urinary metabolome was found to be altered, reflecting the biochemical events associated with HD pathogenesis. The major pathways altered in R6/2 mice were corticosterone dysregulation, insulin resistance, and tryptophan metabolism. The secretion of a pheromone that promotes aggression amongst males while inducing synchronized estrus in females was also blocked. In zQ175DN mice the major variations were associated with oxidative stress biomarkers and tryptophan metabolism pathways. The differences in urinary metabolome reflecting the genetic and phenotypic differences of the two models can help to select the proper one to be used in preclinical studies. Moreover, MS-based metabolomics can help to identify new potential biomarkers of disease progression or efficacy of therapy once validated.

## Figures and Tables

**Figure 1 metabolites-13-00961-f001:**
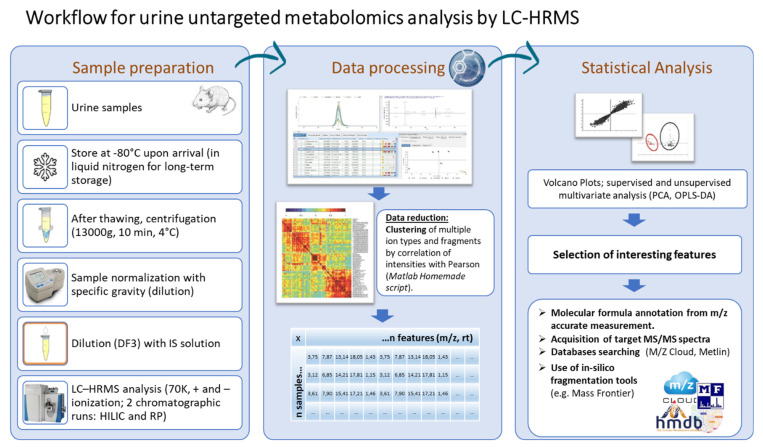
Workflow implemented to explore urine metabolic alteration in R6/2 and zQ175DN HD mice.

**Figure 2 metabolites-13-00961-f002:**
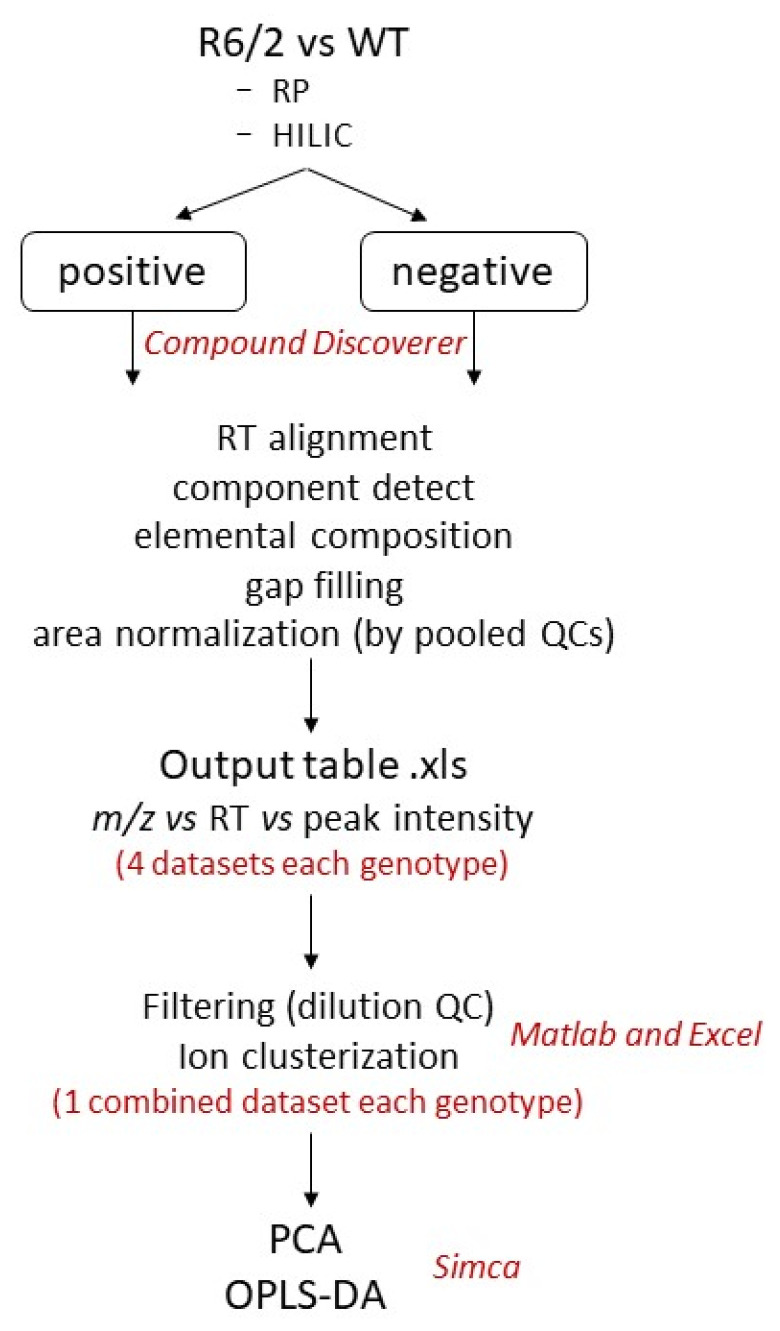
Workflow used for data analysis.

**Figure 3 metabolites-13-00961-f003:**
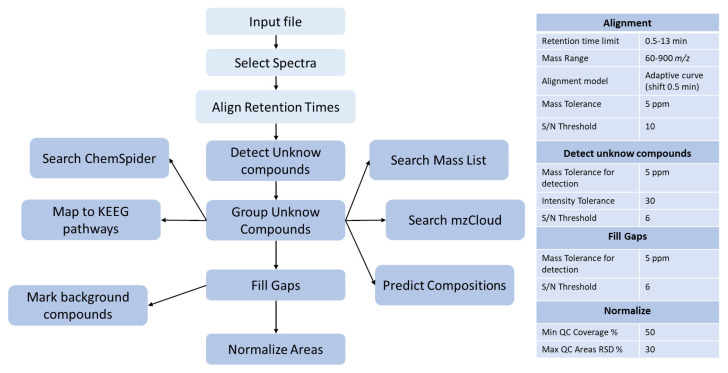
Template used for Compound Discoverer™ data analysis and the principal parameters used.

**Figure 4 metabolites-13-00961-f004:**
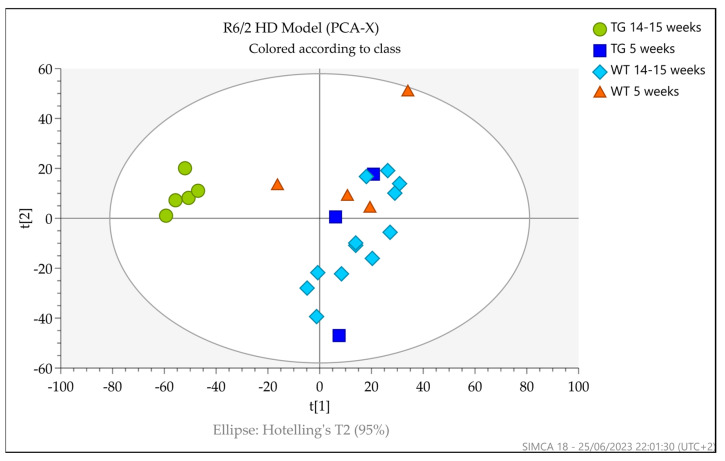
PCA score plot of the filtered dataset (RP and HILIC pos and neg) displaying the separation for R6/2 and wild-type 5- and 14–15-week-old mice. Tg mice at 14–15 weeks (green circle) showed separation from 5-week-old Tg (blue square), 5-week-old non-Tg (red triangle), and 14–15-week-old mice (azure square). One sample behaved as an outlier and was consequently excluded from the OPLS-DA analysis.

**Figure 5 metabolites-13-00961-f005:**
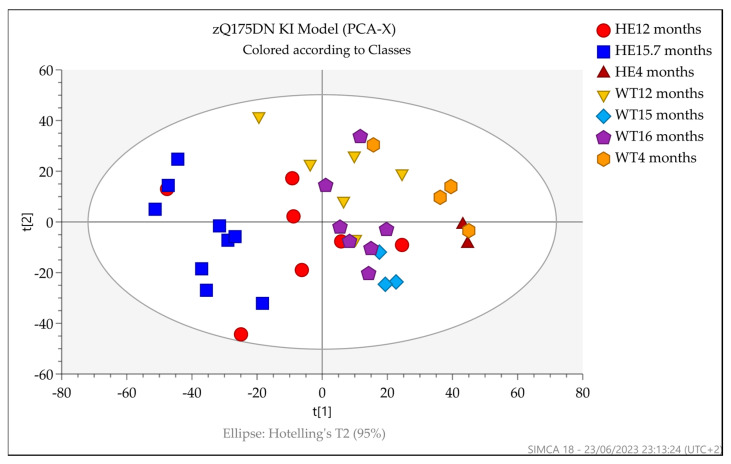
PCA score plot of the filtered dataset (RP and HILIC pos and neg) displaying moderate separation between zQ175DN groups by age and between zQ175DN mice at 15.7 months (blue square) and wild-type groups.

**Figure 6 metabolites-13-00961-f006:**
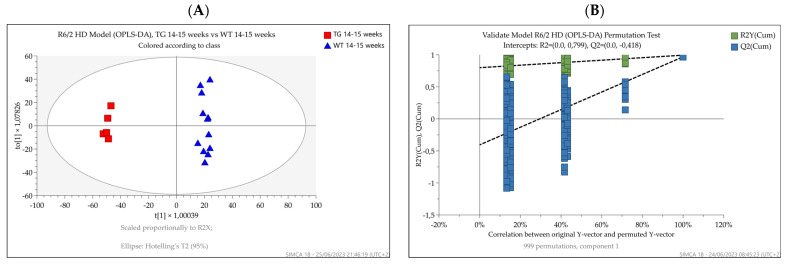
(**A**) OPLS-DA score plots displaying the separation for transgenic R6/2 (red square) and wild-type (blue triangle) 14–15-week-old mice. R2Y and Q2X were 0.99 and 0.97, respectively. (**B**) Permutation test showing the goodness of fit of the model built.

**Figure 7 metabolites-13-00961-f007:**
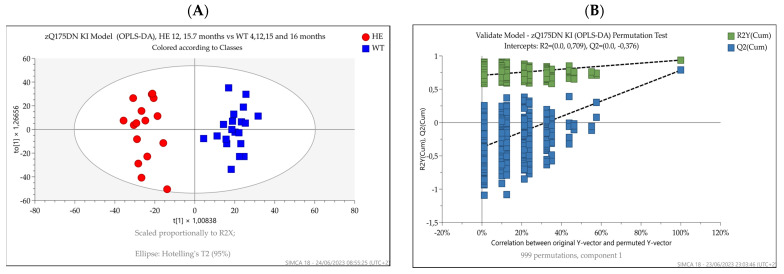
(**A**) OPLS-DA score plots display separation between 12- and 15.7-month-old zQ175DN mice (red circles) and 4-, 12-, and 15.7-month-old wild-type mice (blue squares). R2Y and Q2X were 0.94 and 0.78, respectively. Since only two samples were representing the knock-in mice at four months, they were excluded from the dataset. (**B**) Permutation test showing the goodness of fit of the model built.

**Figure 8 metabolites-13-00961-f008:**
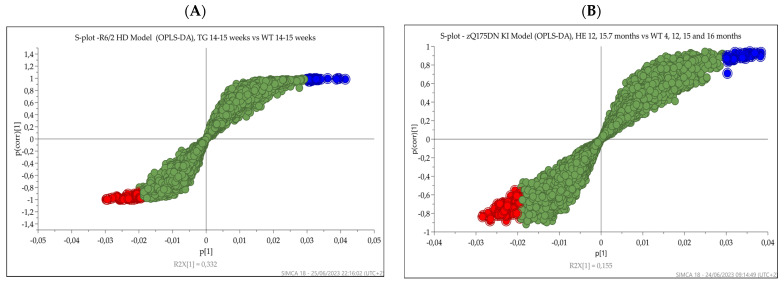
S plot of (**A**) R6/2 versus wild-type mice and (**B**) zQ175DN versus wild-type mice.

**Table 1 metabolites-13-00961-t001:** Altered metabolites in 14–15-week-old R6/2 mice versus age-matched wild-type mice.

RT (min)	Measured Mass	Formula	*p*-Value ^a^	Fold Change ^a^ TG vs. WT	Identification	Pathway
**ESI-**						
7.06	246.9715	C_12_H_8_S_3_	2.00 × 10^−11^	nf in TG	Terthiophene ^b^	unknown
9.24	362.9720	-	2.82 × 10^−14^	nf in TG	unknown	unknown
5.79	422.0610	-	7.93 × 10^−15^	nf in TG	unknown	unknown
8.76	377.1973	C_21_H_30_O_6_	4.68 × 10^−6^	9	18,19-dihydroxycorticosterone ^b^	cortiscosterone dysregulation
9.65	361.2023	C_21_H_30_O_5_	3.82 × 10^−5^	132	18-hydroxycorticosterone ^b^	cortiscosterone dysregulation
9.55	365.2336	C_21_H_34_O_5_	3.05 × 10^−13^	4	Tetrahydrocortisol ^b^	cortiscosterone dysregulation
9.79	363.2180	C_21_H_32_O_5_	2.03 × 10^−5^	8	hydroxy-dehydrotetrahydrocorticosterone ^b^	cortiscosterone dysregulation
9.65	408.2117	C_17_H_32_O_9_N_2_	2.65 × 10^−5^	17	unknown	unknown
**ESI+**						
9.19	379.2112	C_21_H_30_O_6_	2.35 × 10^−6^	218	Hydroxycorticosterone ^b^	cortiscosterone dysregulation
9.95	361.2005	C_21_H_29_O_5_	6.69 × 10^−4^	nf in WT	hydroxy-dehydrocorticosterone ^b^	cortiscosterone dysregulation
6.48	255.1336	C_12_H_18_O_4_N_2_	<1.00 × 10^−4^	10	Pyrraline ^c^	nonenzymatic glycation
2.65	255.1336	C_12_H_18_O_4_N_2_	4.00 × 10^−10^	214	unknown	unknown
4.84	239.1388	C_12_H_18_O_4_N_2_	2.61 × 10^−4^	3.4	unknown	unknown
6.71	144.0841	C_7_H_13_NS	9.27 × 10^−17^	nf in TG	2-(sec-butyl)-4,5-dihydrothiazole ^c^	pheromone
2.49	304.1500	C_12_H_21_O_6_N_3_	2.57 × 10^−13^	44	N-acetyl-aspartyl-lysine ^b^	unknown
6.73	382.0769	C_16_H_15_O_10_N	5.30 × 10^−3^	1.95	xanthurenic acid glucuronide ^b^	Trp metabolism
7.51	206.0448	C_10_H_7_O_4_N	<1.00 × 10^−4^	0.12	xanthurenic acid ^c^	Trp metabolism
8.17	188.0739	C_8_H_14_O_2_NS	2.93 × 10^−6^	0.001	N-[(1E)-3-methyl-1,3-butadien-1-yl]-L-cysteine ^b^	unknown

nf: not found; ^a^
*p*-value and fold change was calculated from the nonparametric Wilcoxon–Mann–Whitney test (one-way ANOVA). ^b^ Putative metabolites were identified by matching the accurate mass, isotope pattern, or MS/MS data with mass spectral and compound libraries. ^c^ Metabolites were confirmed based on the retention time and the MS/MS spectra of synthetic standards.

**Table 2 metabolites-13-00961-t002:** Altered metabolites in 12- and 15.7-month-old zQ175DN versus 4-, 12-, and 15.7-month-old wild-type mice.

RT (min)	Measured Mass	Formula	*p*-Value ^a^	Fold Change ^a^ HE vs. WT	Identification	Pathway
**ESI-**						
4.82	148.9914	C_4_H_5_O_4_S	6.37 × 10^−6^	2.7	thiodiglycolic acid ^b^	oxidative stress
4.43	192.0336	C_6_H_10_O_4_NS	1.11 × 10^−6^	2.5	S-2-carboxyethyl-L-cysteine/ 2-((2-Hydroxyethyl)amino)-3-mercapto-4-oxobutanoic acid ^b^	oxidative stress
4.31	305.0453	C_10_H_13_O_7_N_2_S	9.35 × 10^−6^	nf in WT	deaminated glutathione ^b^	waste compound
9.03	378.0837	C_17_H_16_O_9_N	1.18 × 10^−4^	3.3	indole-3-pyruvic acid glucuronide ^b^	Trp metabolism
8.56	313.0567	C_13_H_13_O_9_	5.97 × 10^−5^	2.9	unknown	Unknown
9.59	435.1303	C_21_H_23_O_10_	4.51 × 10^−4^	3.7	unknown	Unknown
9.56	421.1510	C_21_H_25_O_9_	3.37 × 10^−4^	4.3	unknown	Unknown
9.19	315.1242	C_18_H_19_O_5_	3.29 × 10^−3^	3.9	unknown	Unknown
**ESI+**						
5.10	350.1012	C_12_H_20_O_7_N_3_S	4.00 × 10^−7^	4.4	N-acetylglutathione ^c^	oxidative stress
4.94	398.1663	C_16_H_24_O_7_N_5_	5.62 × 10^−6^	3.9	carnosine-propanal-aspartic ^b^	oxidative stress
1.32	255.0642	C_7_H_15_O_6_N_2_S	2.46 × 10^−7^	1.3	gamma-glutamyltaurine ^b^	dipeptide
1.65	130.1226	C_7_H_16_ON	8.97 × 10^−5^	0.3	N-methyl hexanamide ^b^	fatty amides
0.91	114.0913	C_4_H_8_ON_3_	2.75 × 10^−5^	0.3	creatinine ^c^	others
1.18	116.1069	C_6_H_14_NO	1.53 × 10^−5^	0.3	trimethylaminoacetone ^c^	others

nf: not found; ^a^
*p*-value and fold change was calculated from the nonparametric Wilcoxon–Mann–Whitney test (one-way ANOVA); ^b^ putative metabolites were identified by matching the accurate mass, isotope pattern or MS/MS data with mass spectral and compound libraries; ^c^ metabolites were confirmed with the retention time and the MS/MS spectra of synthetic standards.

## Data Availability

The data presented in this study are available in the article.

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
