# Peer review of "The Urine Metabolome of R6/2 and zQ175DN Huntington’s Disease Mouse Models"

_metabolites, 2023, doi:10.3390/metabo13080961_

Round 1

Reviewer 1 Report

In this manuscript, Speziale and co-workers investigated the alterations of metabolism in urine samples from two widely used, Huntington’s disease mouse models and identified altered levels of some compounds. Moreover, their work revealed that there are differences between these two models. The work is definitely interesting and of significance. The MS analysis is thorough, however, there are some questions and concerns, which need to be addressed by the authors.

Major concerns:

First of all, study design is not optimal. For example, the authors used samples from 3 tg and 4 wild-type 5 weeks-old mice and 5 Tg and 12 wild-type 14-15-weeks-old mice in the case of R6/2 model. It is not really fortunate that there were only 2 4-months old mice for the zQ175DN model, while there were 7 and 9 12-months and 15.7-months old mice, respectively. It would be more useful to use the same mice at different ages, since only urine samples were collected. In that case, disease progression could be compared with the possible fold-change in the identified metabolites, and fewer mice used.

line 146: “The procedure was repeated over several days until 350 µL of urine was collected.” How long did it take exactly?

SG was used for normalization. Since insulin metabolism was found to be affected in one of the mouse models, did the authors check whether there is a correlation between the necessary dilution and the different animal groups?

Do the authors have a plan to identify the unknown metabolites?

Minor concerns:

line 62: What is the neo cassette?

line 81-83: These sentences seem to be rather result/discussion, not introduction.

Please, pay more attention to abbreviations. For example, SG is mentioned at line 151, but it is explained at line 285.

Line 574: Data availability should be stated.

Reviewer 2 Report

After reviewing the manuscript entitled "Urine metabolome of R6/2 and zQ175DN Huntington's disease mouse models", where animal models of Huntington's disease are used, it aims to provide clarity on the molecular mechanisms involved in this neuropathology to facilitate the correct selection of models in preclinical studies, and also provides data on possible specific biomarkers for said neuropathology. In the manuscript, interesting insights have been reported, however the following comments need to be addressed prior to acceptance.

First: Novelty of the manuscript must be better emphasized

Second: Make a scheme of the entire process followed in the experiments with mice and add it in the introduction section.

Third: Please, carefully review the material and method section and try to put original text, most of it is taken literally from other articles. Describe what is new and refer, for the rest of the description, to the manuscript where it was taken. To do this, you can use an anti-plagiarism program that detects these matches

Fourth: The limitations of the study are not noted in the manuscript. Addressing the limitations helps provide a balanced perspective and suggests avenues for future research. Please review and complete the discussion and conclusions, taking this aspect into account.

Minor editing of English language required

Round 2

Reviewer 1 Report

The authors have answered most of my questions, and the manuscript has been significantly improved.

However, I may not be unambiguous considering my comment on the study design. What I did mean that the authors could use for example 5 zQ175DN and 5 control mice, then collect samples from these mice at 4, 12, 15.7-months. This way, it is possible to study alterations as well as these alterations as a function of time and disease progression.  

In the current study: “Since only 2 samples were representing the knock-in mice at 4 months, they were excluded from the dataset.” (line 384). However, in table 2, they show the “Altered metabolites in 4, 12 and 15.7 months zQ175DN versus age matched wild-type mice.” In the table, there are no data concerning the ages of the mice. Were all the alterations observed even at 4-month-old mice? Did the fold change depend on the age of the mice?

Reviewer 2 Report

Authors addressed all the comments and thus enhanced the quality of this manuscript

Author Response

The authors have answered most of my questions, and the manuscript has been significantly improved.

However, I may not be unambiguous considering my comment on the study design. What I did mean that the authors could use for example 5 zQ175DN and 5 control mice, then collect samples from these mice at 4, 12, 15.7-months. This way, it is possible to study alterations as well as these alterations as a function of time and disease progression.  

We thank the Reviewer for clarifying and we apologize for having misunderstood this important suggestion. The Reviewer is perfectly right, longitudinal sample collection would be appropriate and would provide several advantages to obtain very clean data. However, since this was the first urine untargeted metabolome study in HD mice, we were concerned on the long-term storage stability of putative altered metabolites in urine. Especially for the zQ175DN genotype since the symptoms are developed starting from 12 months of age. For this reason, we decided to collect from different age groups, store and process all samples simultaneously, avoiding longer storage for earlier collected samples. It would be very interesting, for further studies, to compare the study design proposed by the Reviewer with the actual one.

In the current study: “Since only 2 samples were representing the knock-in mice at 4 months, they were excluded from the dataset.” (line 384). However, in table 2, they show the “Altered metabolites in 4, 12 and 15.7 months zQ175DN versus age matched wild-type mice.” In the table, there are no data concerning the ages of the mice. Were all the alterations observed even at 4-month-old mice? Did the fold change depend on the age of the mice?

We thank the Reviewer for this comment. At 4 months, urines from zQ175DN were very similar to wild type ones. This is consistent with the slow progression of the disease in zQ175DN mice. However, since we had only two samples from 4 months old zQ174DN mice we decided to not include them in the OPLS-DA.

We have realized that the title of Table 2 is not correct (line 483), we have changed it as follows:

“Table 2. Altered metabolites in 12 and 15.7 months zQ175DN versus 4, 12 and 15.7 months old wild-type mice.”

We have also added to line 383: “12 and 15.7 month’s old”.

We have deleted from line 402: “aged matched”,

and added: “4, 12 and 15.7 months old”.

The fold change reported in Table 2 is the average peak area found in the 12 and 15.7 months old zQ175DN samples versus the average peak area found in the 4, 12 and 15.7 months old wild-type ones. Most metabolites altered at 15.7 months were similarly changed already at 12 months.